# RNA–Binding Protein HuD as a Versatile Factor in Neuronal and Non–Neuronal Systems

**DOI:** 10.3390/biology10050361

**Published:** 2021-04-23

**Authors:** Myeongwoo Jung, Eun Kyung Lee

**Affiliations:** 1Department of Biochemistry, College of Medicine, The Catholic University of Korea, Seoul 06591, Korea; arong1898@catholic.ac.kr; 2Department of Biomedicine & Health Sciences, College of Medicine, The Catholic University of Korea, Seoul 06591, Korea; 3Institute of Aging and Metabolic Diseases, College of Medicine, The Catholic University of Korea, Seoul 06591, Korea

**Keywords:** HuD, RNA–binding protein, neuronal systems, non–neuronal systems, disease pathology

## Abstract

**Simple Summary:**

Tight regulation of gene expression is critical for various biological processes such as proliferation, development, differentiation, and death; its dysregulation is linked to the pathogenesis of diseases. Gene expression is dynamically regulated by numerous factors at DNA, RNA, and protein levels, and RNA binding proteins (RBPs) and non–coding RNAs play important roles in the regulation of RNA metabolisms. RBPs govern a diverse spectrum of RNA metabolism by recognizing and binding to the secondary structure or the certain sequence of target mRNAs, and their malfunctions caused by aberrant expression or mutation are implicated in disease pathology. HuD, an RBP in the human antigen (Hu) family, has been studied as a pivotal regulator of gene expression in neuronal systems; however, accumulating evidence reveals the significance of HuD in non–neuronal systems including certain types of cancer cells or endocrine cells in the lung, pancreas, and adrenal gland. In addition, the abnormal function of HuD suggests its pathological association with neurological disorders, cancers, and diabetes. Thus, this review discusses HuD–mediated gene regulation in neuronal and non–neuronal systems to address how it works to orchestrate gene expression and how its expression is controlled in the stress response of pathogenesis of diseases.

**Abstract:**

HuD (also known as ELAVL4) is an RNA–binding protein belonging to the human antigen (Hu) family that regulates stability, translation, splicing, and adenylation of target mRNAs. Unlike ubiquitously distributed HuR, HuD is only expressed in certain types of tissues, mainly in neuronal systems. Numerous studies have shown that HuD plays essential roles in neuronal development, differentiation, neurogenesis, dendritic maturation, neural plasticity, and synaptic transmission by regulating the metabolism of target mRNAs. However, growing evidence suggests that HuD also functions as a pivotal regulator of gene expression in non–neuronal systems and its malfunction is implicated in disease pathogenesis. Comprehensive knowledge of HuD expression, abundance, molecular targets, and regulatory mechanisms will broaden our understanding of its role as a versatile regulator of gene expression, thus enabling novel treatments for diseases with aberrant HuD expression. This review focuses on recent advances investigating the emerging role of HuD, its molecular mechanisms of target gene regulation, and its disease relevance in both neuronal and non–neuronal systems.

## 1. Introduction

RNA–binding proteins (RBPs) are responsible for the formation of ribonucleoprotein (RNP) complexes by binding to specific sequences or secondary structures of target RNAs. RBPs regulate the life cycle of RNAs, including alternative splicing, maturation, editing, transport, localization, turnover, and translation, thereby acting as an important regulators of gene expression [1,2,3,4]. Canonical RBPs usually include RNA–binding domains (RBDs), such as RNA recognition motif (RRM), K–homology (KH) domains, CCHC–type zinc–finger domains, helicase domains, and glycine–rich domains [5,6]. Non–canonical RBPs lack common RBDs and interact with RNA molecules via intrinsically disordered regions or mono–/di–nucleotide–binding domains [7,8,9]. Mutations or alterations in the expression of certain RBPs are linked to the development of human genetic diseases (reviewed in [10]). Further, impaired expression, mislocalization, and aggregation of RBPs are involved in the pathogenesis of various diseases, such as neurodegeneration, cancer, and metabolic diseases [11,12,13,14,15,16].

HuD, otherwise known as ELAVL4, is an RBP belonging to the Hu/ELAVL (Hu antigen/embryonic lethal, abnormal vision, Drosophila–like) family. While HuR is ubiquitously distributed across tissues, HuD, along with HuB and HuC, exhibits tissue–specific expression, particularly in neurons [17,18,19]. The *HuD* gene generates a variety of mRNA variants through alternative splicing and encodes ~40~42 kDa proteins in humans, mice, and rats [20,21,22]. HuD contains three RRMs (RRM1, RRM2, and RRM3) and a linker region between RRM2 and RRM3, through which it interacts with the AU–rich element (ARE) sequence of target mRNAs, thereby affecting their splicing, polyadenylation, transport, stability, and translation [23].

HuD plays diverse and important roles in neuronal processes, including neuronal development, plasticity, survival, function, and disease processes [23,24,25]. Many studies have emphasized the significance of HuD in the neuronal system; however, it also functions as a pivotal regulator of gene expression in non–neuronal tissues, including lung, testis, pituitary gland, and pancreatic endocrine cells [26,27,28,29]. Therefore, comprehensive knowledge of its expression, abundance, molecular targets, and regulatory mechanisms is needed to broaden our understanding of HuD as a versatile regulator of gene expression. This review focuses on recent studies elucidating the role of HuD, the molecular mechanisms underlying its target gene regulation, and its association with disease in both neuronal and non–neuronal systems.

## 2. General Characteristics of HuD

HuD was identified and characterized as a neuronal form of the Hu family, along with HuB and HuC [17]. HuD is well–conserved among vertebrates and located on chromosomes one, four, and five in humans, mice, and rats, respectively [30,31,32]. The complexity of the 5′ sequence of HuD transcripts and alternative exon splicing may generate several HuD mRNA variants [19,22,33,34] (reviewed in [24]). HuD proteins are ~40~42 kDa in size and have three highly conserved RRMs [23,25]. RRM1 and RRM2 associate with ARE–containing target mRNAs, while RRM3 is known to interact with poly(A) or ARE regions of target mRNAs [35,36,37]. Neuronal Hu proteins HuB, HuC, and HuD share 80% amino acid sequence homology compared to HuR, thus executing a similar role in RNA regulation [19]. The N–terminal and linker regions located between RRM2 and RRM3 seem to be responsible for the Hu family characteristics of each protein, including nuclear–cytoplasmic shuttling, protein–protein interactions, and binding to target mRNAs [24]. HuD variants reportedly display different amino acid sequences at their nuclear localization signal (NLS) or nuclear export signal (NES) in the linker region, which are responsible for temporal and spatial regulation of neuronal differentiation [38].

HuD is primarily found in the brain and regulates neural development, synaptic plasticity, and nerve generation [19,23,27,34]. However, increasing evidence has demonstrated its expression in non–neuronal cells, including small cell lung carcinoma (SCLC), oral squamous cell carcinoma (OSCC), β– and α–cells in the pancreatic islets, thymocytes, cells in the adrenal medulla, and spermatogonial cells in the testis [26,27,28,29,31,39,40,41,42]. This indicates that HuD expression is not ubiquitous, nor is it restricted to neurons and certain types of endocrine cells. To fully understand the cell–type specific roles of HuD, its molecular targets and gene regulatory mechanisms need to be determined.

HuD functions as an important regulator of gene expression by regulating a diverse spectrum of RNA metabolisms, including stability, translation, splicing, polyadenylation, nucleo–cytoplasmic shuttling, and intracellular localization of target mRNAs. HuD increases the stability of target mRNAs by competing with decay factors such as AU binding factor 1 (AUF1) and tristetraprolin (TTP); conversely, it also destabilizes target mRNAs in cooperation with microRNAs [20,24,43,44,45]. Although several studies have shown the role of HuD in mRNA stability, the detailed mechanisms of HuD–mediated regulation of mRNA turnover have not been fully elucidated. Further, HuD can affect translation of target mRNAs in a positive or negative manner. HuD promotes translation of target mRNAs by interacting with eIF4a and poly(A)–binding protein (PABP) [46]. Conversely, HuD functions as a translational repressor by associating with the internal ribosome entry site (IRES) of *p27* mRNA or the stem–loop structure in the 5′UTR of *proinsulin2* (*Ins2*) mRNA [28,47]. In addition to its regulatory role in mRNA turnover and translation, HuD is also involved in post–transcriptional control via exon inclusion or exclusion by splicing, alternative polyadenylation, and site–specific localization of various target mRNAs, thereby contributing to dynamic regulation of gene expression [28,48,49,50,51,52]. In addition, HuD regulates mRNA metabolisms through cooperative interactions with other RBPs including AUF1, insulin–like growth factor 2 mRNA binding protein 1 (IGF2BP1, also known as IMP1 and ZBP1), Ras–GAP SH3 domain binding protein (G3BP), survival of motor neurons (SMN), and PABP [53,54,55,56,57]. A list of target mRNAs and their HuD–mediated regulatory mechanisms is summarized in Table 1.

## 3. Regulation of RNA Metabolism by HuD

Comprehensive understanding of HuD–mediated gene regulation requires identification of its target mRNAs and elucidation of the regulatory mechanisms of RNA metabolism. Several studies have extensively investigated interactions between HuD, its target mRNAs, and HuD–mediated post–transcriptional regulation in neuronal systems, thereby demonstrating the pivotal role of HuD as a neuronal regulator (summarized in Table 1). Additionally, systemic approaches have attempted to identify molecular targets of HuD on a large scale [52,58,59]. For example, HITS–CLIP (high–throughput sequencing of RNA isolated by crosslinking immunoprecipitation) was employed to determine the binding sites targeted by neuronal ELAVLs (nELAVLs), but not specifically HuD, on over 8000 transcripts from the human brain [59]. HuD–associated mRNAs have been analyzed by ribonucleoprotein immunoprecipitation (RIP) followed by microarray in the brains of transgenic mice [58], and also by CRAC (crosslinking and analysis of cDNA) in motor neuron cells expressing His–HA–HuD [52]. Recently, a series of HuD–associating circular RNAs (circRNAs) from HuD transgenic mice were identified by RIP analysis followed by circRNA arrays [60]. These analyses revealed that HuD interacts with a variety of mRNAs as well as non–coding RNAs, via their ARE regions and provided useful information concerning the roles of HuD in RNA regulation. Most studies demonstrated the neuronal function of HuD; however, growing evidence indicates that HuD plays essential roles in HuD–expressing non–neuronal cells, such as pancreatic β–cells and SCLC. In this section, we provide an update on the molecular targets of HuD and HuD–mediated regulatory mechanisms in both neuronal and non–neuronal systems.

### 3.1. Regulation of RNA Metabolism by HuD in Neuronal Systems

Since HuD was first discovered in the brain, its role as an essential regulator governing post–transcriptional control of neuronal gene expression has been extensively reported in drosophila and vertebrates (reviewed in [23,24]). HuD affects diverse neuronal gene expression by regulating mRNA turnover, translation, and splicing. Several studies have demonstrated the role of HuD as a stabilizer of neuronal mRNAs. For example, HuD increases the stability of *growth associated protein 43* (*GAP43*) mRNA in neurons and promotes neurite outgrowth [43,61,62,63,64,65,66,67]. HuD also mediates post–transcriptional control of essential target mRNAs for the brain or neuronal functions, including *brain–derived neurotrophic factor* (*BDNF*) [68,69], *nerve growth factor* (*NGF*) [68], *neurotropin–3* (*NT–3*) [68], *neuro–oncological ventral antigen 1* (*Nova1*) [70], *neuritin 1* [71,72], *neuroserpin* [45], *acetylcholinesterase* (AchE) [73,74], and *special adenine–thymine (AT)–rich DNA–binding protein 1* (*SATB1*) [75], thereby regulating neuronal differentiation, neurogenesis, dendritic maturation, neuronal plasticity, synaptic transmission, and dynamic signaling pathways in neuronal systems. Further, HuD regulates the expression of mRNAs involved in the pathogenesis of neurodegenerative diseases or cancer, including *amyloid precursor protein* (*APP*), *β–site APP–cleaving enzyme 1* (*BACE1*), lncRNA *BACE1AS* [76], *neprilysin* (NEP) [77], *tau* [78,79], *superoxide dismutase 1* (*SOD1*) [80], and *MYCN* [81,82]. In addition to targets found in neuronal tissues, HuD promotes stabilization of target mRNAs also expressed in other tissues, such as *p21* [83], *Ca^2+^/Calmodulin–dependent protein kinase II α* (*CaMKII*α) [84], and *musashi 1* (*MSI1*) [85]. Additionally, an interesting study recently demonstrated that circRNAs, such as *cirHomer1a*, could be molecular targets of HuD [60,86].

With a few exceptions, HuD generally promotes expression of target genes by enhancing translation of their mRNAs. HuD–mediated translational enhancement of *Nova1* [70], *potassium voltage–gated channel subfamily A member 1* (also known as Kv1.1) [87], and several mTORC–responsive genes [52] has been demonstrated in neuronal cells. In addition to turnover and translation of target mRNAs, HuD is also involved in regulating alternative splicing of *calcitonin gene–related peptide* (CGRP) pre–mRNA [48], *neurofibromatosis type 1* (*NF1*) pre–mRNA [49], *APP* mRNA [57], and *glutaminase* mRNA [88].

### 3.2. Regulation of RNA Metabolism by HuD in Non–Neuronal Systems

HuD generally increases the stability of target mRNAs in neuronal cells; however, it decreases the amount of *insulinoma–associated 1* (*INSM1*) mRNA in cooperation with *miR–203a* in pancreatic β–cells [93]. In OSCC cell line HSC3 cells, HuD knockdown downregulated *vascular endothelial growth factor* (*VEGF*)–*A* and –*D*, and *matrix metallopeptidase* (*MMP*)–*2* and –9 mRNAs [39]. These studies determined that target mRNA abundance is altered by HuD knockdown and suggest that HuD regulates mRNA turnover by interacting with decay factors such as microRNAs, AUF1, and TTP in a co–operative or a competitive manner. However, the direct involvement of HuD in the regulation of mRNA stability warrants further investigation.

Several studies have reported that HuD mediates translational control of several target mRNAs in pancreatic β–cells. HuD suppresses translation of *preproinsulin 2* (*Ins2*) mRNA, while enhancing the expression of *preproglucagon* (*Gcg*), *insulin–induced gene 1* (*INSIG1*), *autophagy–related gene 5* (*ATG5*), *p27*, and *mitofusin 2* (*Mfn2*) mRNAs [28,29,89,90,92,94]. These results imply that HuD has a function in the maintenance of glucose homeostasis and β–cell function, and its dysregulation might be involved in the pathogenesis of metabolic diseases such as diabetes.

In addition to translational control of target mRNAs, HuD is also involved in the regulation of splicing and polyadenylation. HuD regulates its own expression by promoting exon 6 inclusion of *HuD* mRNA [91]. Additionally, HuD alters the *Ikaros* (*IK*) isoform profile by regulating alternative splicing of *IK* mRNAs in mouse thymocytes and human T–acute lymphoblastic leukemia (T–ALL) cell line Molt–3 cells in a Notch3–dependent manner [42]. Further, HuD increases the Kv11.1 channel current by affecting alternative polyadenylation of mRNA transcripts of *potassium voltage–gated channel subfamily H member 2* (*KCNH2*), which encodes the Kv11.1 potassium channel [95].

HuD–mediated RNA regulation in non–neuronal cells is relatively unknown compared to that in neuronal systems, but further studies will enable us to explore the specific role of HuD in certain types of cells expressing HuD.

## 4. Disease Relevance of HuD and Its Regulatory Mechanisms

HuD plays important roles in the dynamic regulation of gene expression by affecting RNA metabolism, and its aberrant expression has been reported in several diseases, including neurodegenerative diseases, diabetes, and cancer. Despite the significance of HuD in gene regulation, little is known regarding control of HuD expression in response to stress or the implication of HuD in disease pathogenesis. Herein, we describe the current knowledge of HuD disease relevance as well as the regulatory mechanisms affecting HuD expression, which are summarized in Table 2.

### 4.1. Disease Relevance of HuD

Several studies have implicated HuD in the pathogenesis of neurodegenerative diseases such as Alzheimer’s disease (AD), Parkinson’s disease (PD), and amyotrophic lateral sclerosis (ALS, also known as Lou Gehrig’s disease). Augmented expression of HuD in the brain of patients with AD has been reported [76,96]. Increased expression of HuD may contribute to AD development by increasing expression of mRNAs involved in amyloid–β peptide (Aβ) production, including *APP* and *BACE1* [76]. However, another study reported a reduction of nELAV in the hippocampus of patients with AD and downregulation of HuD after treatment with Aβ42 in human neuroblastoma SH–SY5Y cells [97]. Inconsistent aberrant levels of HuD in AD can be attributed to different brain tissues analyzed between study groups, and further investigation is warranted to clarify the relevance of HuD in AD. In PD, several single–nucleotide polymorphisms (SNPs) were identified in HuD [98,99,100]. Genetic variations in HuD (rs967582, rs2494876, rs3902720) have been associated with age–at–onset (AAO) in PD, while the biological roles of these variations in the regulation of HuD protein abundance or binding affinity to its target mRNAs have not yet been determined. A recent study reported increased levels of HuD proteins in human *induced pluripotent stem cells* (iPSCs) carrying the P525L mutation on the *FUS* gene, which causes ALS [103]. In addition, augmented HuD expression in the motor cortex of patients with sporadic ALS has been associated with superoxide dismutase (SOD) dysregulation [80].

Aberrant expression of HuD also has been determined in various neurological disorders, including epilepsy and schizophrenia. Upregulation of HuD mRNA in the dendritic gyrus after kainic acid–induced seizures and increased dendritic localization of HuD protein in hippocampal neurons, following pilocarpine–induced seizures, were reported in animal models [67,101]. nELAVL null mice displayed the spontaneous epileptic seizure activity resulted from the impaired splicing of genes regulating cellular glutamate level [88]. Additionally, abnormal overexpression of HuD mRNA was observed in the dorsolateral prefrontal cortex of patients with chronic schizophrenia [102]. Although the factors leading to HuD dysregulation or its impact on the regulation of alternative splicing and turnover of target mRNAs are unclear, these reports suggest that abnormal expression of HuD is linked to the pathogenesis of various neurological diseases.

Differential expression of HuD is associated with certain types of cancer, including SCLC, OSCC, neuroblastoma (NB), and pancreatic neuroendocrine tumor (PNET). In patients with SCLC, a more aggressive form of lung cancer, HuD protein was found in patient serum [105,106,107] and HuD mRNA was detected in primary tissues and blood [108,109]. In OSCC, HuD expression was associated with differentiation, metastasis, and invasion of cancer cells, and HuD–positive OSCC cases were associated with a poor survival rate [39]. High HuD mRNA levels were also reported in primary tumor tissues of patients with NB and in several NB cell lines [104,110]. Higher HuD expression was associated with a better clinical outcome in NB, which suggests a role of HuD in decreasing malignancy [104]. Concurring with these results, another study revealed a positive correlation between tumoral HuD loss and significantly reduced survival of patients with PNET. The HuD level was significantly corelated with tumor size and progression of PNET [90]. Although changes in HuD expression in the process of cancer and tumor development are unclear, aberrant HuD levels may provide useful markers for disease diagnosis or prognosis.

In addition to cancer, the disease relevance of HuD has been demonstrated in diabetes, one of the metabolic diseases resulting from impaired glucose homeostasis. Using an animal model of type 2 diabetes mellitus (T2DM), the levels of HuD mRNA and protein were reduced in the pancreas of *db*/*db* mice, suggesting HuD caused β–cell dysfunction [92].

As reported above, HuD is abnormally expressed in several diseases. Although differential regulation of HuD in pathological conditions has not been fully elucidated, understanding the molecular mechanisms fine–tuning HuD expression is critical for therapeutic intervention.

### 4.2. Regulation of HuD Expression

Elucidating the molecular mechanisms modulating HuD expression is required to fully understand how HuD–mediated gene regulation affects RNA metabolism. Several studies have described the regulation of HuD expression by a variety of factors at the transcriptional, post–transcriptional, and post–translational levels.

First, several regulatory mechanisms of HuD transcription have been identified. Neurogenin 2 (Ngn2), the basic helix–loop–helix transcription factor, promotes transcription of HuD by binding to E–boxes in its promoter region, which is essential during neuronal differentiation of P19 cells [22]. SATB1, one of the target mRNAs of HuD, also functions as a transcriptional activator of HuD [75]. Interestingly, HuD and SATB1 cooperatively regulate neural stem and progenitor cell neuronal differentiation via a positive feedback network; HuD stabilizes *SATB1* mRNA, and SATB1 promotes transcription of HuD. Activation of Notch3 signaling contributes to upregulation of HuD expression in thymocytes, which in turn, promotes HuD–mediated splicing of *IK* mRNAs [42]. In mouse pancreatic β–cells, insulin signaling was shown to be responsible for upregulated HuD expression through the IR–IRS–Akt–FoxO1 axis after glucose stimulation [28]. Additionally, thyroid hormone T3 represses transcription of HuD in rat PC12 and mouse N2a cells, and T3 level was inversely correlated with HuD mRNA in the rat brain [111].

Second, HuD expression can be also regulated at the post–transcriptional level. Alternative splicing of HuD mRNAs generate different HuD isoforms exhibiting variable localization patterns, which have been suggested to play different roles in neuronal differentiation and development [22,38]. Neuronal Hu proteins are responsible for exon 6 inclusion of HuD mRNA [91]; however, detailed mechanisms regulating the alternative splicing of HuD mRNA by cis–elements or specific trans–factors have not been fully elucidated. *microRNA–375* (*miR–375*) downregulates HuD expression by destabilizing HuD mRNA and suppressing its translation, thereby affecting neuronal differentiation [27]. *microRNA–129–5p* (*miR–129–5p*) decreases HuD expression and inhibits neurite outgrowth [112]. A recent study reported that RBP Celf1 functions as a translational repressor of HuD during neocortical neurogenesis [113]. Celf1 suppresses translation of HuD mRNA by binding to its 5′UTR region in glutamatergic neurons. Isoform–specific translational repression of HuD mRNAs by Celf1 has been shown to play an important role in neurodevelopment.

Third, HuD protein can be regulated by post–translational modification, including methylation and phosphorylation. Coactivator–associated arginine methyltransferase 1 (CARM1), also known as PRMT4, methylates Arg residues of HuD protein (Arg^236^ in PC12 cells and Arg^248^ residue in MN–1 cells), leading to decreased stability of HuD–mediated *p21* mRNA [114,115]. Methylation of HuD by CARM1 seems to be essential for the transition of neuronal precursor cells from proliferation to differentiation by negatively regulating HuD–mediated gene expression. In addition to methylation, phosphorylation of neuronal Hu proteins by protein kinase C (PKC) has been reported [67,76,115]. PKCα induces phosphorylation of the Thr residue in neuronal Hu proteins, which in turn, promotes *GAP–43* mRNA stabilization in SH–SY5Y cells [116]. PKC contributes to neuronal differentiation by affecting HuD–mediated RNA metabolism, which directly regulates binding between HuD and target mRNAs, or by regulating factors that affect HuD functions, such as CARM1, in an indirect manner [68,77].

As described above, several factors regulating HuD expression have been identified (Figure 1), but the detailed mechanisms of regulation warrant further investigation. Additional studies examining specific regulators directing HuD abundance or activity are expected to provide novel insights to facilitate the development of treatments for diseases caused by HuD malfunction.

## 5. Concluding Remarks and Perspectives

RBPs function as critical effectors of gene expression and their malfunctions are implicated in disease pathology, including RBP gene mutations, altered RBP expression, and aggregation and sequestration of RBPs with RNAs or other proteins. Therefore, approaches that restore the abundance or function of RBPs have great potential for clinical applications [10,117]. HuD, an RBP in the Hu family, is a versatile protein that regulates various aspects of RNA metabolism, including splicing, stability, and translation of target mRNAs, and is therefore involved in various cellular processes, including cell growth, apoptosis, differentiation, and metabolism. The majority of studies have focused on the role of HuD in neuronal systems; however, accumulating evidence indicates that HuD is also expressed in non–neuronal cells, such as pancreatic β–cells, thymocytes, and SCLC, and its differential expression is implicated in the pathogenesis of several diseases. In this review, we summarized the current knowledge of molecular targets, disease relevance, and regulatory mechanisms of HuD.

Despite continued studies elucidating HuD–mediated gene regulatory mechanisms, several questions remain unanswered. Which characteristics of HuD are distinct from those of other Hu proteins? What mechanism contributes to cell–type specific expression and function of HuD? What signals or cellular conditions regulate HuD expression at the transcriptional, post–transcriptional, and post–translational level during disease development? Besides HuD abundance, which mechanisms determine subcellular localization and binding affinity to its interacting partners? What signals or stimuli affect the binding of HuD to target mRNA? What mechanisms determine competitive or cooperative association between HuD, miRNA, and other RBPs on target mRNAs? Addressing these questions based on systemic and/or integrated approaches using multi–omics analysis will enhance our knowledge of HuD–mediated gene regulation.

Although neuronal and non–neuronal cells express HuD, we still do not know how the detailed mechanisms regulating HuD expression or HuD–mediated RNA regulation are different among cell types. What are the common characteristics of HuD–expressing cells? Do common signaling pathways direct HuD expression in neuronal and non–neuronal systems or not? Do both systems have a common mechanism or cell type–specific mechanisms in mRNA regulation? Further studies enable us to fully explore the gene networks regulated by HuD, thus improving our understanding of diseases associated with aberrant HuD expression for therapeutic intervention.

## Figures and Tables

**Figure 1 biology-10-00361-f001:**
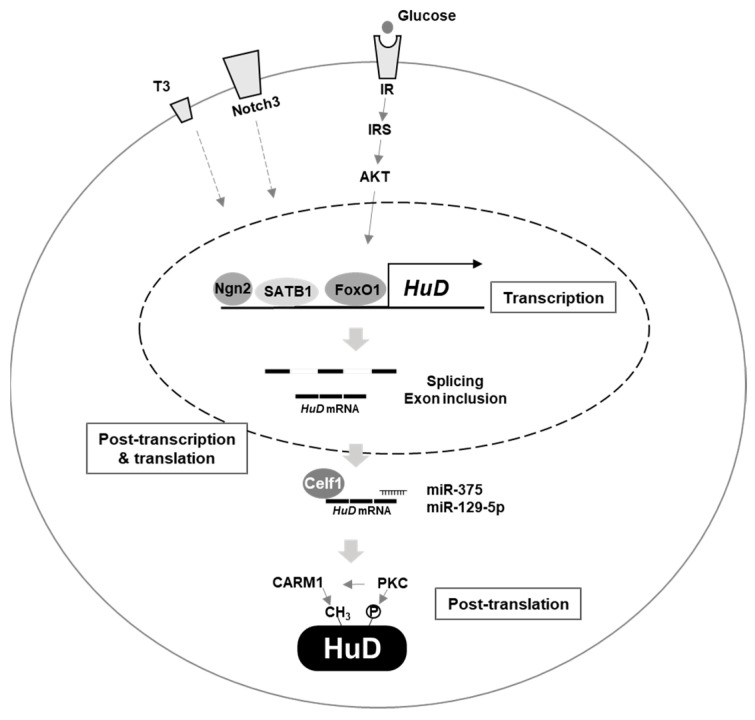
Regulation of HuD expression. Several factors affect HuD expression. Ngn2, SATB1, IR/IRS/AKT/FoxO1, notch3, and thyroid hormone T3 regulate transcription of HuD gene. Celf1, miR–375, miR–129–5p and alternative splicing are involved in post–transcriptional and translational control of HuD mRNA. CARM1 and PKC mediate post–translational regulation of HuD protein.

**Table 1 biology-10-00361-t001:** Target RNAs and their regulatory mechanisms.

Target	Study Systems	Regulatory Mechanism	Function	Ref.
**I. Neuronal cells or brain**				
*Acetylcholinesterase* (*AChE*)	Rat pheochromocytoma–derived cell PC12 Superior cervical ganglion (SCG) from rat Brain from HuD O/E mice	mRNA stability ↑		[73,74]
*Amyloid Precursor Protein* (*APP*)	Human neuroblastoma SK–N–F1 Brain from HuD O/E mice Brain from AD patient	mRNA stability ↑	APP → Aβ processing ↑	[76]
	Human neuroblastoma SK–N–SH	Alternative splicing(Exon 7 and 8 exclusion ↑)		[57]
*β–site APP–cleaving enzyme 1* (*BACE1*) and *BACE–AS*	Human neuroblastoma SK–N–F1 Brain from HuD O/E mice Brain from AD patient	mRNA stability ↑	APP → Aβ processing ↑	[76]
*Brain Derived Neurotrophic Factor* (*BDNF*) long 3’UTR	Hippocampal neuron from E18 rat Hippocampal, cortical neuron from E17 mice Mouse catecholaminergic neural tumor cell CAD Brain from HuD O/E mice	mRNA stability ↑	Dendritic maturation ↑	[68,69]
*Calcitonin Gene–Related Peptide* (*CGPR*) pre–mRNA	Human cervical tumor Hela Chinese hamster ovary (CHO) cell Mouse testicular teratoma F9 Rat pheochromocytoma–derived cell PC12 Human neuroblastoma SK–N–SH Mouse teratocarcinoma P19 Rat medullary thyroid carcinoma CA77	Alternative splicing (Exon4 exclusion ↑)		[48]
*Calcium/Calmodulin Dependent Protein Kinase II Alpha* (*CaMKⅡ**α*)	Hippocampal neuron from E18–19 rat	mRNA stability ↑		[84]
*CDKN1A* (*p21*)	Rat pheochromocytoma–derived cell PC12	mRNA stability ↑	Cell proliferation ↓	[83]
*circHomer protein homolog1a* (*cirHomer1a*)	Brain (frontal cortex) from HuD K/O and O/E mice	Synaptic expression ↑		[60,86]
*Growth Associated Protein 43* (*GAP–43*)	Rat pheochromocytoma–derived cell PC12 Mouse embryonic stem cell AB2.2 Cortical neuron from E19 rat Rat DRG/mouse neuroblastoma hybrid cell F11 Brain from rat Brain from HuD K/O and O/E mice	mRNA stability ↑ Transportation into neurites ↑	Neurite outgrowth ↑	[43,54,61,62,63,64,65,66,67]
*Glutaminase* (*Gls*)	Brain (cortex) from HuC, HuD double K/O mice	Alternative splicing (Gls–long isoform ↓)		[88]
*MYCN*	Human neuroblastoma NBL–W–N Mouse fibroblast NIH 3T3	mRNA stability ↑		[81,82]
*Neprilysin* (*NEP*)	Human neuroblastoma SK–N–SH	mRNA stability ↑	Aβ levels ↓ by NEP	[77]
*Nerve Growth Factor* (*NGF*)	Hippocampal neuron from E18 rat	mRNA stability ↑	Dendritic maturation ↑	[68]
*Neuritin 1* (*Nrn1/Cpg15*)	Rat pheochromocytoma–derived cell PC12 Human neuroblastoma SH–SY5Y DRG neuron from rat Cortical neuron from E18 rat Hippocampal neuron from E18 rat Rat DRG/mouse neuroblastoma hybrid cell F11 Brain from HuD KO mice	Axonal localization ↑ mRNA stability ↑		[55,71,72]
*Neurofibromatosis type 1* (*NF–1*) pre–mRNA	Human cervical tumor cell Hela Rat medullary thyroid carcinoma CA77 Mouse embryonic stem cell R1 Cerebellar neurons from mice	Alternative splicing (Exon23a skipping ↑) Local transcription rate ↑ (NF–1 gene exon 23a)		[49,51]
*Neuroserpin*	Brain from rat Rat pheochromocytoma–derived cell PC12	mRNA stability ↑ (?)		[45]
*Neurotrophin 3* (*NT–3*)	Hippocampal neuron from E18 rat	mRNA stability ↑	Dendritic maturation ↑	[68]
*NOVA Alternative Splicing Regulator 1* (*NOVA–1*)	Mouse motor neuronal cell NSC34	mRNA stability ↑ Translation ↑	Splicing activity	[70]
*Musashi–1* (*MSI1*)	Neural stem/progenitor cell (NSC) in SVZfrom mice Human neuroblastoma SH–SY5Y	mRNA stability ↑		[85]
*Potassium voltage–gated channel subfamily A member 1* (*Kv1.1*)	Cortical neuron from E18–19 rat	Translation ↑		[87]
*Special AT–rich DNA–binding protein 1* (*SATB1*)	Neural stem/progenitor cell (NSC) in SVZ from HuD KO mice	mRNA stability ↑	NSC differentiation ↑	[75]
*Superoxide Dismutase 1* (*SOD1*) long 3′UTR	Human neuroblastoma SH–SY5Y Brain from ALS patients	mRNA stability ↑		[80]
*Tau*	Rat pheochromocytoma–derived cell PC12Mouse teratocarcinoma P19	Transportation into neurites ↑	Neurite outgrowth ↑	[53,78,79]
Others: mTORC–responsive genes	Mouse motor neuronal cell NSC34	Translation ↑		[52]
**II. Non–neuronal cells or other tissues**				
*Autophagy Related Gene 5* (*ATG5*)	Mouse insulinoma βTC6 Pancreatic islet from HuD KO mice, *db/db* mice	Translation ↑	Autophagosome formation ↑	[89]
*CDKN1B* (*p27*)	Human embryonic kidney cell 293T and human cervical tumor Hela Mouse insulinoma βTC6 and MIN6 Pancreatic NET from patients	Translation ↑ or ↓	Cell proliferation ↑ or ↓	[47,90]
*HuD* mRNA	Human cervical tumor HelaRat medullary thyroid carcinoma CA77	Alternative splicing(Exon 6 inclusion ↑)		[91]
*Hu Antigen R* (*HuR*)	Mouse teratocarcinoma P19	Alternative polyadenylation ↑		[50]
*Insulin Induced Gene 1* (*Insig1*)	Mouse insulinoma βTC6	Translation ↑	TG accumulation ↓	[92]
*Insulinoma–Associated Protein 1* (*INSM1*)	Mouse insulinoma βTC6	mRNA stability ↓		[93]
*Ikaros* (*IK*)	Thymocyte from N3–Ictg, N3–Ictg/pTα^−/−^ and pTα^–/–^ mice Human T–All cell line Molt–3	Alternative splicing (Ik–6, 8, 5/7, 9 ↑)	T cell lymphomagenesis	[42]
*Matrix Metallopeptidase–2* and *–9* (*MMP–2* and *–9*)	Human oral squamous cell carcinoma HSC3	mRNA stability ↑ (?)		[39]
*Mitofusin 2* (*Mfn2*)	Mouse insulinoma βTC6Pancreatic islet from HuD KO mice		Mitochondria fusion ↑	[94]
*Potassium Voltage–Gated Channel Subfamily H Member 2* (*KCNH2*)	Human embryonic kidney 293	Alternative polyadenylation ↓	Kv11.1a isoform expression ↑ Kv11.1 channel current ↑	[95]
*Preproglucagon* (*Gcg*)	Mouse glucagonoma αTC1 Pancreatic islet from HuD KO mice	Translation ↑	Glucagon biosynthesis	[29]
*Preproinsulin2* (*Ins2*)	Mouse insulinoma βTC6 Pancreatic islet from HuD KO mice	Translation ↓	Insulin biosynthesis	[28,56]
*Vascular Endothelial Growth Factor–A* and *–D* (*VEGF–A* and *VEGF–D*)	Human oral squamous cell carcinoma HSC3	mRNA stability ↑ (?)		[39]

↑ means its upregulation (increase). ↓ means its downregulation (decrease). → means from APP to Aβ.

**Table 2 biology-10-00361-t002:** Disease relevance of HuD.

Disease	Disease Relevance of HuD	Ref.
Alzheimer’s diseases (AD)	*HuD* mRNA and HuD protein ↑ in superior temporal gyrus (STG) of AD patients	[76]
	HuD protein ↑ in the brain of AD patients	[96]
	nELAVL protein ↓ in hippocampus of AD patients	[97]
Parkinson’s diseases (PD)	Several SNPs (rs967582, 2494876, 3902720) were identified.	[98,99,100]
Epilepsy	*HuD* mRNA↑ in dentate gyrus of kainic acid–induced seizures model	[67]
	Dendritic localization of HuD protein ↑ in hippocampal neurons of pilocarpine–induced seizure model	[101]
Schizophrenia	*HuD* mRNA ↑ in the dorsolateral prefrontal cortex from patients with chronic schizophrenia	[102]
Amyotrophic lateral sclerosis (ALS)	*HuD* mRNA and HuD protein ↑ in motor cortex of sporadic ALS patients	[80]
	HuD protein ↑ in human iPSCs carrying the FUS^P525L^ mutation	[103]
Neuroblastoma	*HuD* mRNA was detected in primary NB tumor samples.	[104]
Small cell lung carcinoma (SCLC)	HuD protein ↑ in serum from SCLC patients	[105,106,107]
	*HuD* mRNA ↑ in primary tissue from SCLC patients	[108]
	*HuD* mRNA ↑ in blood from SCLC patients	[109]
Oral squamous cell carcinoma (OSCC)	HuD (+) group is associated with poor prognosis of OSCC patients.	[39]
Pancreatic neuroendocrine tumor (PNET)	HuD (–) group is associated with poor prognosis of PNET patients.	[90]
Type 2 diabetes mellitus (T2DM)	*HuD* mRNA and HuD protein ↓ in islet from *db/db* mice	[94]

↑ means its upregulation (increase). ↓ means its downregulation (decrease).

## Data Availability

Not applicable.

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
