# Peer review of "RNA–Binding Protein HuD as a Versatile Factor in Neuronal and Non–Neuronal Systems"

_biology, 2021, doi:10.3390/biology10050361_

Round 1
Reviewer 1 Report
In this review article, authors have described regulation of HuD at the transcription and translation levels as well as HuD’s role as a RNA binding protein. HuD was originally identified as an important RBP during neurogenesis and maintenance of terminally differentiated neurons. More recently the role of HuD has become apparent in non-neuronal tissues such as some cancers and pancreatic islets.
In some sections more explanation or clarification is required (see below):
1) Page 2 – Lines 63 & 64: Cis-acting region specificity for 3 RRMs is described. What is the difference in binding to 3 RRMs? Do all 3 RRMs interact simultaneously or is there any specificity depending upon the target mRNA?
2) Page 3 – Line 112: Can authors describe what they mean by “via various RNA regions”? On the previous page, authors indicated that HuD RRMs interact with polyA or ARE. It is important to identify whether binding to mRNAs and ncRNAs occurs through similar cis-acting regions.
3) Page 10 – Line 15 from bottom of the page: Please check if STAB1 or SATB1 promotes HuD transcription.
4) Page 10 – Line 3 from bottom of the page: accidently authors stated HuD pre-mRNA. It should read HuD mRNA.
5) Authors should include their own thoughts as to what is their opinion about HuD role in neurogenesis and non-neuronal tissues.
Author Response
1) Page 2 – Lines 63 & 64: Cis-acting region specificity for 3 RRMs is described. What is the difference in binding to 3 RRMs? Do all 3 RRMs interact simultaneously or is there any specificity depending upon the target mRNA?
--> Biochemical investigation using deletion mutants of HuD and reporter genes have shown a reduction in binding activity of HuD to target mRNAs, which suggests that all three RRMs are important to binding ability of HuD. RRM 1 and 2 have a predominant role in binding of HuD to ARE sequence on target mRNAs, while RRM3 plays a role in the formation of a stable RNP complex by interacting with other proteins (NAR 1997, 25:3564-9 and MCB 2000, 20:4765). However, whether three RRMs of HuD can generally act on target mRNAs is needed to be determined.
2) Page 3 – Line 112: Can authors describe what they mean by “via various RNA regions”? On the previous page, authors indicated that HuD RRMs interact with polyA or ARE. It is important to identify whether binding to mRNAs and ncRNAs occurs through similar cis-acting regions.
--> The authors appreciate this comment.
HuD is known to associate with its target mRNAs by binding to ARE regions. Recent studies show that HuD also binds to the ARE regions on circular RNAs (Front Genetics 2020, 11:790, Mol Psychiatry 2020, 25:2712). We revised manuscript as the Reviewer suggested.
3) Page 10 – Line 15 from bottom of the page: Please check if STAB1 or SATB1 promotes HuD transcription.
--> The authors appreciate this comment. We corrected it in revised manuscript.
4) Page 10 – Line 3 from bottom of the page: accidently authors stated HuD pre-mRNA. It should read HuD mRNA.
--> The authors appreciate this comment. We corrected it in revised manuscript.
5) Authors should include their own thoughts as to what is their opinion about HuD role in neurogenesis and non-neuronal tissues.
--> The authors appreciate this comment. We revised the text as the Reviewer advised.
Reviewer 2 Report
I read the review by Jung and colleague with much interest. The authors presented a well-written review of RNA-binding protein HuD in neuronal system, neural-associated diseases as well as other systems and pathologic conditions including cancer. Tables and figure are exhaustive and nicely summarize the major messages. I have some minor corrections and suggestions:
1) Please, check numbers of sections and subsections: the authors went from paragraph number 2 to paragraph 3.1. Please check and correct.
2) Page 2, line 44: should be “RRM3”
3) Page 3, line 144: should be “translation”
4) The authors should consider to add miR-129-5p among the microRNAs regulating HuD expression, based on this recent publication: Loffreda Alessia et al., miR-129-5p: A key factor and therapeutic target in amyotrophic lateral sclerosis. Progress in Neurobiology. 2020. In case, please update Figure 1 accordingly.
5) Interactions among different RNA binding proteins is not uncommon. The family of RNA binding proteins IGF2BP (also known as IMP) is involved in neural development as well as various pathologic conditions like cancer. If evidences are available in literature, the authors should consider to discuss direct interactions between HuD and other RNA binding proteins, for instance the members of the IGF2BP family. This could further enrich the section 1 concerning General characteristics of HuD.
Author Response
I read the review by Jung and colleague with much interest. The authors presented a well-written review of RNA-binding protein HuD in neuronal system, neural-associated diseases as well as other systems and pathologic conditions.
-->The authors are grateful for the positive reviews, and for valuable comments to improve the manuscript.
I have some minor corrections and suggestions:
1) Please, check numbers of sections and subsections: the authors went from paragraph number 2 to paragraph 3.1. Please check and correct.
--> The authors appreciate this comment. We corrected it in revised manuscript.
--> The authors appreciate this comment. We corrected it in revised manuscript.
3) Page 3, line 144: should be “translation”
--> The authors appreciate this comment. We corrected it in revised manuscript.
4) The authors should consider to add miR-129-5p among the microRNAs regulating HuD expression, based on this recent publication: Loffreda Alessia et al., miR-129-5p: A key factor and therapeutic target in amyotrophic lateral sclerosis. Progress in Neurobiology. 2020. In case, please update Figure 1 accordingly.
--> The authors thank the Reviewer for this suggestion. We updated the text and Figure 1 as the Reviewer suggested.
5) Interactions among different RNA binding proteins is not uncommon. The family of RNA binding proteins IGF2BP (also known as IMP) is involved in neural development as well as various pathologic conditions like cancer. If evidences are available in literature, the authors should consider to discuss direct interactions between HuD and other RNA binding proteins, for instance the members of the IGF2BP family. This could further enrich the section 1 concerning General characteristics of HuD.
--> The authors are grateful for valuable suggestion to improve the manuscript. We revised the manuscript by including this information.
Reviewer 3 Report
Overall, this is a balanced, comprehensive review of RNA-binding protein HuD in neuronal and non-neuronal systems.
I have few comments:
What are the differences in HUD expression and mRNA metabolism in neuronal vs non neuronal systems? Are there any specific molecular pathways which dominates more in neuronal vs non neuronal cells?
What are the potential mechanisms by which HUD regulates mRNA turnover?
Author Response
Overall, this is a balanced, comprehensive review of RNA-binding protein HuD in neuronal and non-neuronal systems.
--> The authors are grateful for the positive reviews, and for valuable comments to improve the manuscript.
I have few comments:
What are the differences in HUD expression and mRNA metabolism in neuronal vs non neuronal systems? Are there any specific molecular pathways which dominates more in neuronal vs non neuronal cells?
--> Detailed mechanisms regulating HuD expression in neuronal and non-neuronal systems are not fully elucidated.
Some regulators have been identified; Ngn2, SATB1, and T3 (at the transcription level in neuronal cells), miR-375, miR-129-5p, and Celf1 (at the posttranscriptional level in neuronal cells), and glucose and FoxO1 (at transcriptional level in pancreatic β-cells), as described in Figure 1. However, comparative studies are not investigated whether these regulations are specific to the neuronal cells or non-neuronal cells. Although our unpublished results show certain DAMP (damage-associated molecular pattern) molecules directly down-regulate HuD expression in both neuronal cells and pancreatic β-cells, further studies are required to fully understand how HuD expression is controlled in both systems. HuD regulates mRNA metabolism with a mechanism common, but may regulate mRNA metabolism by other mechanisms, such as having unique interacting partners or associating to the cell type-specific mRNAs (insulin and glucagon mRNAs) in neuronal and non-neuronal cells. Since there have not been many comparative studies among cell types on the HuD expression or regulatory mechanism of HuD so far, we need to expand our knowledge to fully understand the roles of HuD.
We revised the text to include the above.
What are the potential mechanisms by which HUD regulates mRNA turnover?
--> HuD may affect to mRNA turnover 1) by competing with decay factors that destabilizing target mRNAs (microRNA, mRNA destabilizing proteins such as AUF1, CNOT1, TTP) or 2) by facilitating the interaction between target mRNAs and decay factors in a cooperative way, as described in the text. Some of studies have shown HuD-mediated mRNA turnover, however, further studies are still needed to elucidate the detailed mechanism of it.